# *Cryptococcus neoformans* Prp8 Intein: An In Vivo Target-Based Drug Screening System in *Saccharomyces cerevisiae* to Identify Protein Splicing Inhibitors and Explore Its Dynamics

**DOI:** 10.3390/jof8080846

**Published:** 2022-08-12

**Authors:** José Alex Lourenço Fernandes, Matheus da Silva Zatti, Thales Domingos Arantes, Maria Fernanda Bezerra de Souza, Mariana Marchi Santoni, Danuza Rossi, Cleslei Fernando Zanelli, Xiang-Qin Liu, Eduardo Bagagli, Raquel Cordeiro Theodoro

**Affiliations:** 1Institute of Tropical Medicine, Federal University of Rio Grande do Norte (UFRN), Natal 59077-080, Rio Grande do Norte, Brazil; 2Department of Biochemistry, Biosciences Center, Federal University of Rio Grande do Norte (UFRN), Natal 59078-900, Rio Grande do Norte, Brazil; 3Ottawa Hospital Research Institute (OHRI), The University of Ottawa, Ottawa, ON K1H 8M5, Canada; 4Department of Biosciences and Technology, Institute of Tropical Pathology and Public Health, Federal University of Goiás, Goiânia 74605-050, Goiás, Brazil; 5School of Pharmaceutical Sciences, São Paulo State University (UNESP), Araraquara 14800-903, São Paulo, Brazil; 6Pensabio, São Paulo 05005-010, São Paulo, Brazil; 7Department of Biochemistry and Molecular Biology, Dalhousie University, Halifax, NS B3H 4R2, Canada; 8Microbiology and Immunology Department, Biosciences Institute of Botucatu, São Paulo State University (UNESP), Botucatu 18618-689, São Paulo, Brazil

**Keywords:** protein splicing, intein, Prp8, Ura3, CnePrp8i, cisplatin, novel drug target, inhibitor screening

## Abstract

Inteins are genetic mobile elements that are inserted within protein-coding genes, which are usually housekeeping genes. They are transcribed and translated along with the host gene, then catalyze their own splicing out of the host protein, which assumes its functional conformation thereafter. As Prp8 inteins are found in several important fungal pathogens and are absent in mammals, they are considered potential therapeutic targets since inhibiting their splicing would selectively block the maturation of fungal proteins. We developed a target-based drug screening system to evaluate the splicing of Prp8 intein from the yeast pathogen *Cryptococcus neoformans* (CnePrp8i) using *Saccharomyces cerevisiae* Ura3 as a non-native host protein. In our heterologous system, intein splicing preserved the full functionality of Ura3. To validate the system for drug screening, we examined cisplatin, which has been described as an intein splicing inhibitor. By using our system, new potential protein splicing inhibitors may be identified and used, in the future, as a new class of drugs for mycosis treatment. Our system also greatly facilitates the visualization of CnePrp8i splicing dynamics in vivo.

## 1. Introduction

Fungal diseases kill more than 1.5 million people per year and more than one billion individuals are currently living with a fungal infection [1]. In this scenario, cryptococcosis is considered one of the most important systemic mycoses, affecting both immunocompetent and immunocompromised individuals. It is caused by *Cryptococcus* spp., whose propagules are inhaled from the environment, firstly affecting the lungs and, in most cases, the central nervous system. The infection rapidly evolves and may lead to death by cryptococcal meningitis. Immunocompromised individuals, such as HIV patients, are frequently affected [2]. It is estimated that approximately one million cases of cryptococcal meningitis occur among HIV-positive individuals, which results in approximately 624,700 deaths in the first three months after infection [3]. A recent study showed that pathogenic *Cryptococcus* species kill between 180,000 and 220,000 people each year [4,5], making cryptococcosis the sixth most fatal infectious disease, behind only COVID-19, AIDS, tuberculosis, malaria and deaths associated with diarrhea, according to the World Heathy Organization [6,7,8,9]. In addition, outbreaks of cryptococcosis in immunocompetent individuals can be observed, such as the one in Vancouver, British Columbia, Canada [10,11,12]. There are two pathogenic species complexes in the *Cryptococcus* genus, *Cryptococcus neoformans* and *Cryptococcus gattii*, comprising seven cryptic species [13]. The *C. neoformans* complex is responsible for 72.17% of all *Cryptococcus* infections and it is globally endemic [14].

Despite the important impact of cryptococcosis and other mycoses on public health, only three classes of antifungals are available and drug-resistant strains are notoriously frequent [15,16]. Indeed, drug resistance is of sufficient risk to public health that the World Health Organization recognized it as being one of the top 10 global public health threats facing humanity [17]. The available antifungal drugs can cause serious side effects, such as hepatotoxicity and nephrotoxicity, especially polyene and imidazole derivatives [18,19,20,21,22]. As fungal and mammalian cells are closely related [23], specific drug targets are not easily found. In this sense, inteins (internal proteins), a class of mobile genetic element, seem to be good candidates for such a task, as (i) they are present in many pathogenic fungi, but absent in the genome of metazoan organisms; (ii) they are integrated into important proteins for cell function; (iii) their correct splicing is indispensable for the host protein′s function and (iv) they have a highly evolutionarily conserved splicing mechanism.

Inteins are intervening protein-coding sequences that are transcribed and translated along with the host protein (extein) and then undergo an autocatalytic protein splicing, releasing themselves from the host protein, which forms the mature functional protein. The mobility of some inteins, known as bi-functional or full-length inteins, is achieved via a homing endonuclease (HE) domain located between their N- and C-terminal splicing domains. These inteins are widespread and can be found in all domains of life as well as in viruses [24,25,26,27]. Inteins usually occur in conserved sites of housekeeping genes, which are essential for the survival of the host organism [28]. It is believed that the low mutational rates of housekeeping genes preserve HE recognition sites, while genes with high mutational rates would rapidly become immune to the HE [29].

For many years, inteins were thought to be exclusively parasitic due to the HE domain, responsible for their rapid spread and fixation in populations. However, recently this idea has been challenged by pieces of evidence showing that inteins from *sufB* and *radA* genes, from *Mycobacterium tuberculosis* and *Pyrococcus horikoshii*, respectively, can act as post-translational regulators [30,31]. Additionally, inteins can mediate the production of new proteins by protein alternative splicing (iPAS) [32]. These new data have taken us to a new level of intein understanding. Nowadays, some inteins are hypothesized to be gene expression modulators that may lead to a greater protein diversity without any modification at the genetic level.

In *Cryptococcus* pathogenic species, mini-inteins (inteins without the HE domain) are found within the pre-mRNA-splicing factor 8 (Prp8) protein, encoded by the *PRP8* gene. Prp8 is one of the largest and most conserved nuclear proteins, and plays a central role in the catalytic portion of the spliceosome, and participates in several molecular rearrangements for nuclear pre-mRNA intron splicing, an essential step in gene expression. Therefore, Prp8 is indispensable for cell survival [33].

The Prp8 intein (Prp8i) is sporadically distributed in all fungal phyla [24,34] and includes pathogenic species that cause systemic mycoses, such as *Histoplasma capsulatum*, *Paracoccidioides brasiliensis*, *Paracoccidioides lutzii*, *Emmonsia parva*, *Blastomyces dermatitidis*, *C. neoformans* and *C. gattii*, and also dermatophytes, such as *Microsporum* spp. and *Trichophyton* spp. [35,36,37,38,39,40]. The inhibition of Prp8 intein splicing, resulting in a non-functional host protein, has been proposed as a new possible approach for antifungal treatment [41,42,43]. In vivo and also in vitro screening systems for such inhibition were created mainly for RecA intein in an attempt to screen possible antituberculosis compounds [44,45,46,47] and, recently, an in vitro screening system based on green fluorescent protein was applied for evaluating the inhibition of Prp8 intein from *C. neoformans* and *C. gattii* (CnePrp8i and CgaPrp8i, respectively) by compounds from a small-molecule library [48], however, still, in vivo systems for evaluating CnePrp8i splicing have yet to be developed.

Taking into account the challenging treatment of cryptococcal infection, we developed an in vivo target-based drug screening system in *Saccharomyces cerevisiae* for CnePRP8i splicing evaluation using the Ura3 protein as a non-native extein. Our results indicate that the CnePRP8i performs its splicing from Ura3 protein and therefore this system is suitable for selection or validation of potential splicing inhibitors by culturing *S. cerevisiae* in medium without uracil and 5-FOA medium. This system could serve as a tool for the identification of novel antifungals.

## 2. Materials and Methods

### 2.1. System Design and Work

To evaluate the CnePrp8i protein splicing in our yeast heterologous system, we used the W303 strain of *S. cerevisiae*, pRS313 plasmid and different Ura3 protein constructs containing the CnePRP8i to evaluate its splicing (Appendix A). The decision to use Ura3 as a host protein was based on the well-characterized and widely available biochemical tools to evaluate its function. Ura3 catalyzes the conversion of orotidine-5′-phosphate (OMP) into uridine monophosphate (UMP), being the sixth step in the de novo biosynthesis of pyrimidines; Ura3 can also convert the non-toxic 5-fluoroorotic acid (5-FOA) into 5-fluorouracil, which is a toxic compound. Using those principles, we constructed a system that may be subject to positive and negative selection. For the former, growth is related to intein splicing and, for the latter, growth is related to the absence of splicing. If intein splicing occurs, Ura3 returns to its original and functional conformation; if the splicing is inhibited or blocked, the CnePrp8i will not be released from Ura3, resulting in a non-functional protein. Protein splicing could be assessed by growth in three different culture media: (i) synthetic complete without histidine (SC -His), (ii) synthetic complete without histidine and uracil (SC -His -Ura) and (iii) 5-FOA (Figure 1).

### 2.2. Cloning, Site-Directed Mutagenesis and S. cerevisiae Transformation

The *S. cerevisiae* Ura3 protein sequence was manually analyzed and advantageous possible intein insertion sites were identified by observing three key points: (i) biochemical similarities to the intein native insertion site, (ii) location in a non-rigid region (i.e., loop regions) and (iii) sites not directly related to the protein function. The desired coding sequence and controls were synthesized by GenScript (Piscataway, NJ, USA) and provided in the pUC57 plasmid. To subclone the sequences to pRS313, which is suitable for *S. cerevisiae*, both plasmids were digested with BamHI and EcoRI (New England Biolabs, Ipswich, MA, USA). The desired generated fragments were purified, ligated with T4 DNA Ligase (New England Biolabs, Ipswich, MA, USA), transformed into chemically competent *Escherichia coli* DH5α and then grown at 37 °C in solid Luria–Bertani (LB) medium (1% tryptone, 1% sodium chloride, 0.5% yeast extract, 2% agar) supplemented with ampicillin (100 μg/mL) for 16 h. Positive colonies were screened by PCR (using both M13 and pCnePRP8i primer sets—Appendix A), plasmid digestion and DNA sequencing, which was carried out at UNESP (Botucatu, São Paulo, Brazil) in an ABI3500 sequencer (Applied Biosystems, Foster City, CA, USA).

The subcloning of constructs synthesized by GenScript (full plasmid sequences and their maps are available in Appendix A) from pUC57 to pRS313, as well as the site-directed mutagenesis to create a construct in which the intein would be unable to splice out, were confirmed in three different ways: PCR of the bacterial colony with primers M13, digestion with XhoI and sequencing analysis. Our constructs were named as pRHis, pRExt, pRInt and pRMut, to describe, respectively, the pRS313 plasmid (i) with 6×his-tag and *URA3* gene; (ii) with 6×his-tag and *URA3* gene containing codons for the amino acid residues from the PRP8 gene that, in the native extein, flanks the CnePrp8i; (iii) with 6×his-tag and *URA3* gene having Prp8 amino acid residues flanking the CnePrp8i; (iv) with 6×his-tag and *URA3* gene having Prp8 amino acid residues flanking a mutated and splicing incompetent version of CnePrp8i (Figure 2A). The confirmed plasmids were delivered to *S. cerevisiae* using standard protocols [49]. All transformed yeast cells with pRS313 were expected to grow in SC -His medium; cells with functional Ura3 (with pRHis, pRExt and pRInt) would grow in the SC -His -Ura medium; and only cells with non-functional Ura3 (with pRMut) would grow on the 5-FOA medium (Figure 2B). The amino acid modifications introduced in Ura3 and the amino acid sequences of all constructs are available in Appendix A.

#### 2.2.1. DNA and RNA Manipulation

Plasmids were transformed and maintained in *E. coli* and, whenever needed, extracted using the GeneJET Plasmid Miniprep Kit (Thermo Fisher Scientific, Carlsbad, CA, USA). *S. cerevisiae* total RNA was extracted using TRIzol reagent, PureLink RNA Mini Kit and PureLink DNase set (Thermo Fisher Scientific, Carlsbad, CA, USA). The RNA extraction quantification and purity were assessed by NanoDrop 2000 (Thermo Fisher Scientific, Wilmington, DE, USA). cDNA synthesis was performed by using the RevertAid RT Reverse Transcription Kit (Thermo Fisher Scientific, Carlsbad, CA, USA). All the necessary PCR reactions were performed with Phusion High-Fidelity DNA Polymerase (New England Biolabs, Ipswich, MA, USA) with 1× CG buffer, 3% DMSO or 1.5 M betaine, 200 μM deoxynucleoside triphosphates, 1 μM each primer, 30 ng template DNA, 0.02 U/μL Phusion DNA Polymerase (New England Biolabs, Ipswich, MA, USA). PCR was carried out with an initial denaturation step at 98 °C for 3 min, followed by 40 cycles of denaturation at 98 °C for 30 s, annealing temperature required for each primer set (Appendix A) for 30 s, and extension at 72 °C for 30 s/kb and then a final extension at 72 °C for 10 min. Amplicons were separated by 1% agarose electrophoresis gel stained with ethidium bromide at 90 V for 90 min. When necessary, PCR product bands were sliced and purified with the Illustra GFX PCR DNA and Gel Band Purification Kit (GE Healthcare Life Sciences, Chicago, IL, USA) and used for downstream applications. The manufacturer′s instructions were always followed.

#### 2.2.2. Site-Directed Mutagenesis by Inverse PCR

To create an intein unable to perform splicing, the two essential amino acid residues (cysteine and asparagine) for the splicing were substituted for alanine by site-directed mutagenesis by inverse PCR. The first mutation was incorporated by using the primer set ppRInt(C→A) and the PCR reaction, amplicon manipulation, ligation and *E. coli* transformation were carried out as previously mentioned. Once the plasmid with the first mutation was established, the second mutation was incorporated by using the primer set ppRInt(N→A). Positively transformed colonies were screened by PCR (using both M13 and pCnePRP8i primer sets), plasmid digestion and DNA sequencing in Myleus Facilities (Belo Horizonte, Minas Gerais, Brazil).

#### 2.2.3. *S. cerevisiae* Manipulation

*S. cerevisiae* strain W303 was obtained from the Laboratory of Molecular and Cellular Biology of Microorganisms, School of Pharmaceutical Sciences, UNESP, Araraquara, São Paulo, Brazil. Standard methods were used for its growth and manipulation. Before the transformation, yeast was cultured in YPD (1% yeast extract, 2% Gibco™ Bacto™ peptone and 2% glucose) and then transformed by the LiAC/SS-DNA/PEG method with all the constructed plasmids. Positively transformed yeast cells were selected by synthetic complete medium without histidine (SC -His-0.67% yeast nitrogen base without amino acids, 2% glucose, 0.077% drop-out supplement without histidine) and confirmed by PCR. For Ura3 activity positive selection, transforming yeasts were screened by their ability to grow in synthetic complete medium without histidine and uracil (SC -His -Ura) (0.67% yeast nitrogen base without amino acids, 2% glucose, 0.075% drop-out supplement without histidine and uracil), by producing endogenous uracil. For Ura3 activity negative selection, transforming yeast clones were screened by their toxicity to 5-fluoroorotic acid medium (5-FOA-0.67% yeast nitrogen base without amino acids, 2% glucose, 0.1% 5-fluoroorotic acid, 50 μg/mL uracil, 0.075% drop-out without histidine and uracil). Yeast cultures were incubated at both 25 °C and 37 °C.

### 2.3. Protein Analysis

Yeast cells were grown in appropriate media (SC -His or SC -His -Ura) until they reached OD_600_ ~1.0; cells were then mechanically disrupted (using acid-washed 425–600 μm glass beads and a FastPrep FP120 cell disrupter) in lysis buffer (50 mM Tris-EDTA, pH 7.5; 100 mM KCl; 1 mM dithiothreitol; and complete protease inhibitor cocktail tablets). Protein extracts were quantified by the Bradford method; 70 μg of each protein extract were separated by SDS-PAGE under reducing conditions and transferred to a nitrocellulose membrane. Membranes were blocked with 5% skim milk powder in TBS-T (10 mM Tris-HCl; pH 7.5; 150 mM NaCl and 0.05% Tween) and incubated at 4 °C overnight with the mouse-derived anti-6×his-tag monoclonal primary antibody or the rabbit-derived anti-eiF5A polyclonal primary antibody. Secondary antibodies for chemiluminescent detection were horseradish peroxidase (HRP)-conjugated rabbit antimouse and goat antirabbit IgG. Chemiluminescence was generated using the ECL Prime Western Blotting System (GE Healthcare, Little Chalfont, Buckinghamshire, UK) and digitally captured using a charge-coupled device (CCD) camera in a C-Digit Blot imaging system (Li-Cor). The relative semiquantitative protein ratio (Ura3/eiF5a) was determined by ImageJ 1.52a [50].

### 2.4. Bioinformatic Analysis

Previously generated *S. cerevisiae* label-free total proteome sequencing data were assessed by PaxDB [51]. Only high-quality proteomes covering both eiF5a and Ura3 were kept for later use. The Ura3/eiF5a ratio was then determined using LibreOffice Calc.

### 2.5. Cisplatin Assay

Cisplatin interference in the intein splicing was measured according to the up-to-date ‘Antifungal MIC method for yeasts′ protocol established by the European Committee on Antimicrobial Susceptibility Testing (EUCAST) [52]. Cisplatin (Sigma) was serially diluted in 11 different concentrations in 100 μL of ddH_2_O. The final tested concentrations ranged from 4.12 μM to 4216.53 μM. All the three media were tested (SC -His, SC -His -Ura and 5-FOA). As cisplatin is photosensitive, all the experiments were conducted in the dark. Growth was measured by OD_530_ after 24 h and 48 h. Biological triplicates were performed, and growth in the presence of drug was normalized against inoculation controls (without drug). Non-inoculation controls were used for each tested concentration and therefore used as blanks.

One-way ANOVA (analysis of variance) followed by post hoc comparison analysis using Tukey′s honestly significant difference (HSD) test was performed using the Bioinfokit Toolkit [53] and statsmodels module [54] in Python 3.8.6 [55]. To ensure that the data were drawn from normal distribution and the treatments have homogeneity of variances, we used the Shapiro–Wilk test and Bartlett’s test, respectively. Data were plotted and finalized in Inkscape [56].

## 3. Results

### 3.1. Yeast Growth Pattern (Temperature Sensitivity)

Saturated cultures of W303 yeast cells transformed with each construct subcloned on the pR313 plasmid were plated in 6-fold serial dilutions and their growth was assessed independently at 25 °C and 37 °C on SC -His, SC -His -Ura and 5-FOA media. The temperature did not change the yeasts′ growth pattern and intein splicing was reconstituted on pRInt, making functional copies of Ura3 (Figure 3).

### 3.2. Assessment of CnePrp8i Presence at mRNA Level

A reverse transcription, followed by PCR, using primers for the *URA3* coding sequence (primer set p3s and p102s), confirmed the transcription of all the constructs, as well as ruled out any intein splicing event at the RNA level. As expected, intein-less controls exhibited an 845 bp PCR product (Figure 4A—lanes 4 and 5) and CnePrp8i-containing yeast showed a 1400 bp PCR product (Figure 4A—lanes 6 and 7), demonstrating that the CnePrp8i splicing was not happening at the RNA level. As intermediary PCR products were also obtained for CnePrp8i-containing constructs (Figure 4A—lanes 6 and 7), we then performed a PCR using CnePrp8i-specific primers, which gave us 509 bp amplicons only for CnePrp8i-containing constructs (Figure 4B—lanes 6 and 7).

To determine whether our samples had genomic or plasmid DNA contamination or not, we also performed another PCR using the primer set p35 and p102. As expected, no amplification was observed (data not shown).

### 3.3. Evaluation of CnePrp8i Splicing by Western Blot

We used anti-6×his-tag antibody to perform Ura3-6×his-tag protein detection by Western blot using protein extracts from yeast cells transformed with pRInt and pRExt and grown in SC -His -Ura media (Figure 5A); a thin band was also observed in yeast cells transformed with pRInt plasmid grown in SC -His media, although the band was faint (Figure 5A—lane 4). Comparing cell growth on the same media and using eIF5A as a loading control, it was observed that the Ura3 quantities produced by pRExt and pRInt transformed cells were quite similar, while pRHis transformed cells produced less Ura3 (Figure 5B). Available public protein mass spectrometry data analysis also supports our findings by showing that the Ura3/eIF5A ratio increases when uracil-less media is used (Figure 5B—shaded area).

### 3.4. Test of Cisplatin in the Proposed In Vivo System

Cisplatin has been proposed to be a splicing inhibitor for MtuRecAi and also CnePrp8i and CgaPrp8i in in vitro screening systems [57,58]. For this reason, we tested this drug to assess its influence on the CnePrp8i splicing in our in vivo system. Transformed *S. cerevisiae* strains were grown on appropriate media in eleven serial concentrations of cisplatin (ranging from 4216.52 μM to 4.11 μM). Regardless of the construct, medium and time, cisplatin is toxic to *S. cerevisiae* cells, exhibiting an IC_50_ of 201.65 μM.

All major differences between the pRInt and its controls were found at a concentration of 131.76 μM cisplatin after 24 h of growth (Figure 6A) and at 263.53 μM after 48 h (Figure 6B). The control constructs pRHis and pRExt showed a highly similar growth pattern, while pRInt exhibited significantly impaired growth at both 24 h and 48 h. However, this difference between pRInt and its controls was, unexpectedly, observed in both SC -His and SC -His -Ura media. Since yeast cultures do not need endogenous uracil in SC -His medium, the lower growth of pRInt yeasts cannot be directly associated with splicing inhibition and Ura3 impairment. These results indicated that the simple presence of the intein seems to increase cisplatin toxicity.

## 4. Discussion

The Prp8 intein found in human pathogenic fungi is a promising drug target as it is located in one of the most conserved genes among eukaryotes. If intein splicing could be inhibited pharmacologically, the host Prp8 would no longer be functional, interrupting the survival and proliferation of the fungal cell. Since both the Prp8i insertion site and the flanking amino acid residues necessary for splicing are conserved among fungal pathogens [58], the discovery of a drug that inhibits this intein splicing could have a broad-spectrum antifungal activity.

For the first time, a eukaryotic in vivo screening system for splicing was proposed for a fungal intein. Other authors previously assessed the Prp8i splicing activity in a non-native host protein (maltose binding protein and thioredoxin) in *E. coli* for the following species: *Aspergillus fumigatus*, *A. nidulans*, *H. capsulatum*, *C. neoformans*, *P. brasiliensis*, *P. lutzii*, *B. dermatitidis* and *E. parva* [39,41]. Unfortunately, Western blots were needed to assess the splicing, which makes the system very laborious to be applied in any drug screening system. By inserting the CnePrp8i in the *URA3* gene, we aimed to create a screening drug system based on two approaches: positive selection (growth in a medium lacking uracil is associated with splicing) and negative selection (growth is associated with the absence of splicing in 5-FOA media).

Despite the robustness of intein splicing evidenced in numerous non-native models, the main challenge of this work was to insert the Prp8i into a Ura3 site that would be similar to its native context. Any necessary alterations of the selected insertion site should not change the Ura3 functionality, since we focused our strategy on Ura3 activity in order to relate intein splicing to an easily identifiable phenotype.

After obtaining all the constructs, cloning and obtaining the transformed *S. cerevisiae*, we characterized the growth pattern of each construct strain, and examined how temperature influences the intein splicing in our system. We also demonstrated that the added 6×his-tag does not influence Ura3 activity. We additionally showed that all the residue modifications in Ura3, which were necessary to recapitulate the same splicing junctions found in Prp8i’s native context (i.e., the extein Prp8 protein), did not affect Ura3 enzyme activity. Yeast cells transformed with the pRInt exhibited growth in SC -His -Ura, but not in 5-FOA, which indicates that the CnePrp8i released itself from Ura3 and that Ura3 functionality was reestablished. Ura3 is a dimeric protein and the insertion site was within the edge of the dimer interface (Appendix A). For pRMut, in which the activity of the splicing was interrupted, there was no growth in SC -His -Ura.

Other important validation steps were to confirm that the constructs were successfully transcribed and to ensure that the intein was not removed at the RNA level. By RT-PCR, we showed that all the constructs were expressed and that, in our system, CnePrp8i was not spliced at the RNA level. Faint PCR bands could be observed even in non-transformant yeast cells. Actually, the W303 strain has a defective nuclear copy of the *URA3* gene with a transition at nucleotide 701 from G to A, changing amino acid 234—a key amino acid for Ura3 interaction with its ligand [59]—from glycine to glutamate [60,61]. The faint band of pRInt and pRMut, together with the *URA3-PRP8i* (the larger band), in the same sample, indicates that the endogenous mutated *URA3* gene is less transcribed than its plasmid counterpart.

The next step was to validate by Western blot the obtained results for growth. We showed that *URA3* expression is only detected in the absence of uracil in the medium and this finding is in agreement with all the research about *URA3* regulation, which demonstrates that its expression could be increased by more than five-fold in the absence of uracil [62,63]. There were no data regarding the variations at the protein level, but our analysis of publicly available proteomics data (also normalized by eIF5A) showed that Ura3 protein abundance is higher when yeast cells are cultivated in uracil-deprived media. No precursor protein (when the intein remains ligated at its extein) was detected, which also demonstrates our system’s robustness in terms of intein splicing. On the other hand, a limitation of our system was the inability to detect the Ura3-Prp8i-mut (pRMut) by Western blot, which could be explained by the fact that (i) in SC -His -Ura medium, yeast strains with this construct do not grow (as in Figure 3) and (ii) in SC -His medium, Ura3 expression is low (since there is uracil in the medium), as evidenced in Figure 5B. The use of a stronger promoter would perhaps overcome this.

Systems developed for high-throughput drug screening are essential as they isolate the target (in this case, the intein) and maximize its exposure to a great number of different drugs. However, in in vivo applications, the target is not isolated and the drug is exposed to a wide range of possible enzymatic modifications that may make its use unfeasible. Likewise, off-target effects are common. Such effects may explain the great number of false positives found in high-throughput screening systems [64].

The search for intein splicing inhibitors is not recent and metal molecules, such as zinc and copper, were initially described as reversible inhibitors [65,66]. This discovery allowed splicing modulation and the development of in vitro systems for high-throughput screening, initially for prokaryotic inteins, with a special focus on their antimycobacterial potential [45,46,57,67]. According to Green et al. (2019) [68], Cu^2+^ reversibly modifies the catalytically active cysteine from the N-splicing domain, while Zn^2+^ binds at the cysteine and also at the C-terminal asparagine, inhibiting the splicing of the Prp8 intein. The authors also reported non-metal molecules such as reactive nitrogen species compounds as splicing inhibitors.

In a high-throughput screening analysis carried out by Zhang et al. (2011) [57,69], approximately 50 compounds, among them cisplatin, acted as MtuRecA intein splicing inhibitors, with IC_50_ in the micromolar range. Unfortunately, the majority of the identified inhibitors were electrophiles, which may exhibit cytotoxicity and are not suitable for long-term therapy. However, the employment of improved techniques for chemical synthesis, molecular changes and the use of new drug delivery systems may permit a decrease in toxicity of some compounds. Thus, an in vivo system which enables an easy validation of data obtained by in vitro studies is extremely important.

Cisplatin was also shown to inhibit CnePrp8i and CgaPrp8i splicing in an in vitro splicing assay based on split *Renilla* luciferase with an IC_50_ and of 2.5 µM [58]. Obviously, in in vivo systems, this concentration needs to be increased. For example, in the present study, 131.76 μM of cisplatin was necessary for growth inhibition of *S. cerevisiae* containing the CnePrp8i in Ura3, after 24 h. This inhibitory concentration was also larger than that obtained in in vivo systems for splicing inhibition of MtuRecAi, whose IC_50_ is around 10 μM. This difference may be related to biochemical and physiological differences between bacteria (*E. coli* was used in this in vivo assay) and yeast, especially when considering drug-exporting proteins in yeast [70,71].

Our results showed clearly that the presence of CnePrp8i in the *URA3* gene increased the cytotoxicity of cisplatin, but we are not able to conclude that this was only due to splicing inhibition as, even when grown in an uracil-supplemented medium (SC -His), yeast strains transformed with pRInt exhibited decreased growth. The mechanism for inhibition of the Prp8 intein splicing was very well detailed previously, revealing that residues C1, T61, H62, H64, H169, N170 and D95 are essential for complex formation. For example, if C1 and H169 codons are mutated, cisplatin cannot bind to the intein [58]. Therefore, we speculate that the ligation of Prp8i to cisplatin, besides preventing intein splicing, could dam cisplatin molecules inside the yeast cell. This might occur even when yeast cells do not need uracil (in SC -His medium) as there is a basal *URA3* gene expression in this condition.

In vivo screening systems for intein splicing evaluation, such as the one developed here, are not only important for the selection or validation of drugs that may inhibit intein splicing, but also for a better understanding of the splicing dynamics and physiological factors that may interfere with its efficiency, as recent studies have pointed out that inteins may modulate gene expression at the post-translational level.

## Figures and Tables

**Figure 1 jof-08-00846-f001:**
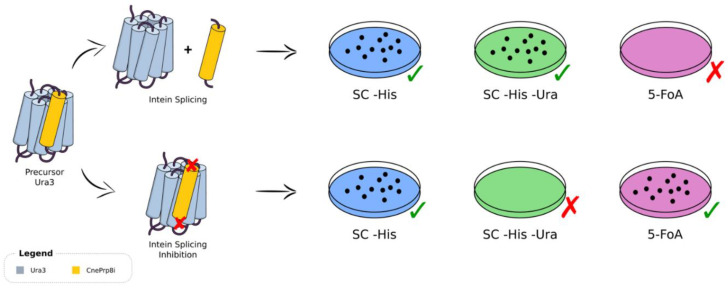
An overview of the system. Intein splicing might occur normally (top) or be inhibited (bottom), which would be visualized by the different growth patterns generated in given media (left side panel). SC -His: synthetic complete medium without histidine; SC -His -Ura: synthetic complete medium without histidine and uracil; 5-FOA: SC -His -Ura medium with 5-fluoroorotic acid and uracil.

**Figure 2 jof-08-00846-f002:**
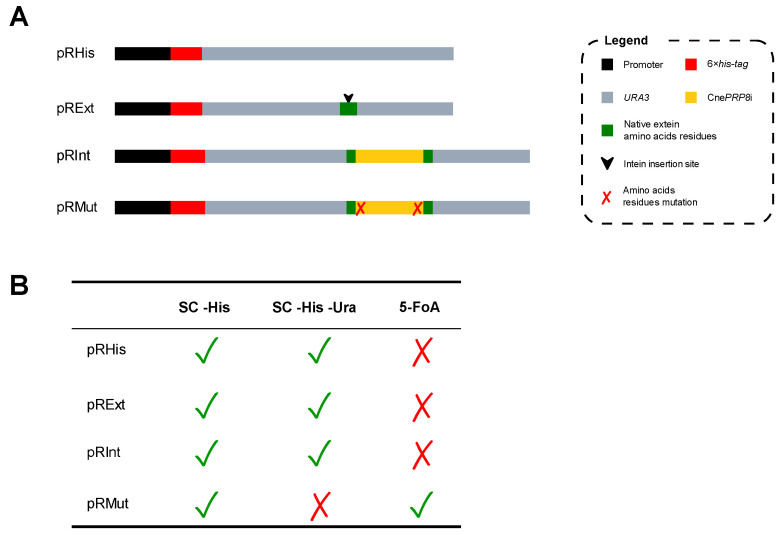
Gene construct overview and growth pattern in different culture media. (**A**) Gene construct maps highlighting their promoters and 6×his-tag positions, including all the necessary alterations. (**B**) Expected growth pattern for each construct under normal conditions.

**Figure 3 jof-08-00846-f003:**
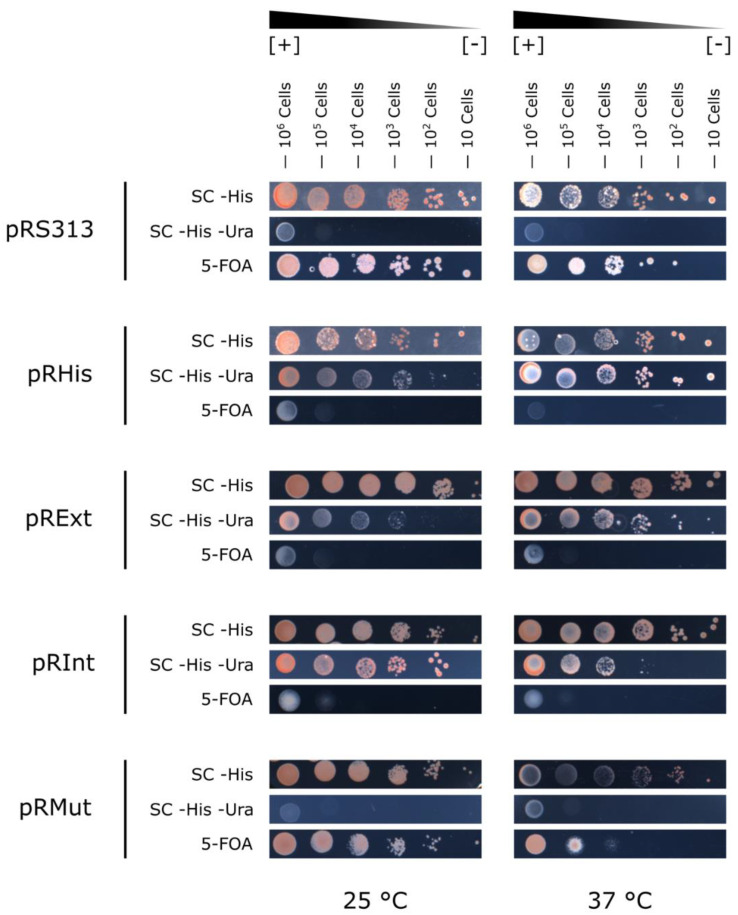
CnePrp8i splicing assessed by growth pattern of pRS313, pRHis, pRExt, pRInt and pRMut transformed yeast cells. Using the yeast spot plating assay, 10^6^ to 10 yeast cells were cultured on SC -His, SC -His -Ura and 5-FOA media at 25 °C and 37 °C and the growth pattern was according to the expected in Figure 2B.

**Figure 4 jof-08-00846-f004:**
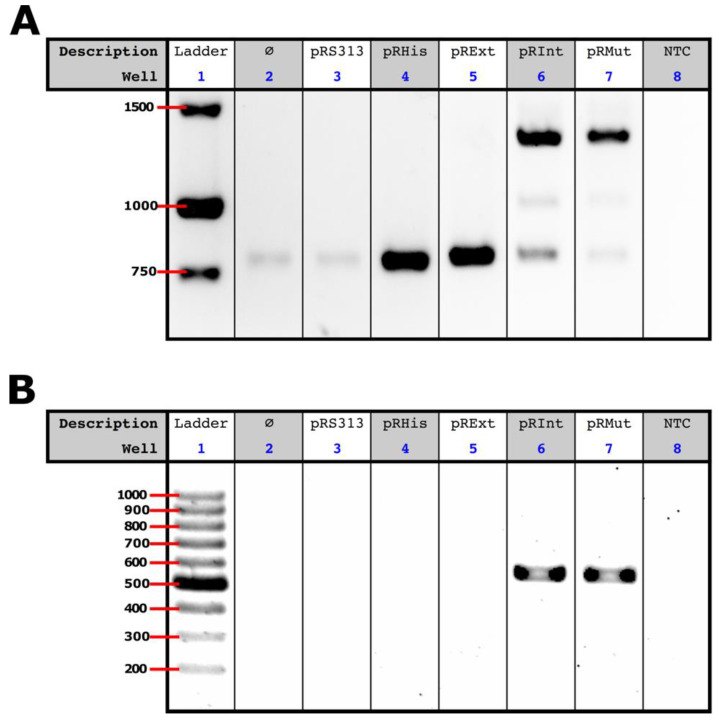
RT-PCR for evaluation of CnePrp8i presence at the RNA level. (**A**) Gel electrophoresis of RT-PCR products for the full *URA3* coding sequence; and (**B**) for the CnePrp8i coding sequence. The empty symbol (∅) stands for non-transformed yeast cells and NTC stands for non-template control.

**Figure 5 jof-08-00846-f005:**
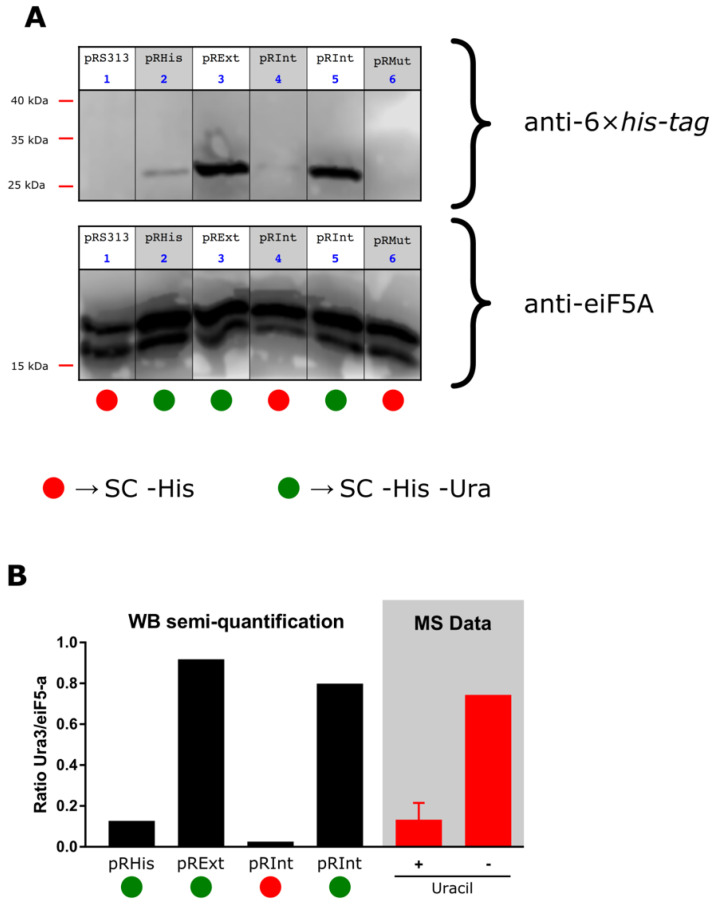
Western blots and semi-quantification of Ura3 protein levels. (**A**) Western blot using an anti-6×his-tag antibody, showing the Ura3-tagged protein (top), and anti-eiF5A, used as a loading control (bottom). (**B**) Relative Ura3 semi-quantification compared to its respective control; in the gray area, the same ratio was assessed using mass spectrometry (MS)-based proteomics data under two different environments: high uracil availability (+) and low uracil availability (−) media. Western blot gels were edited to exclude unnecessary lanes; however, scale and visualized bands were preserved.

**Figure 6 jof-08-00846-f006:**
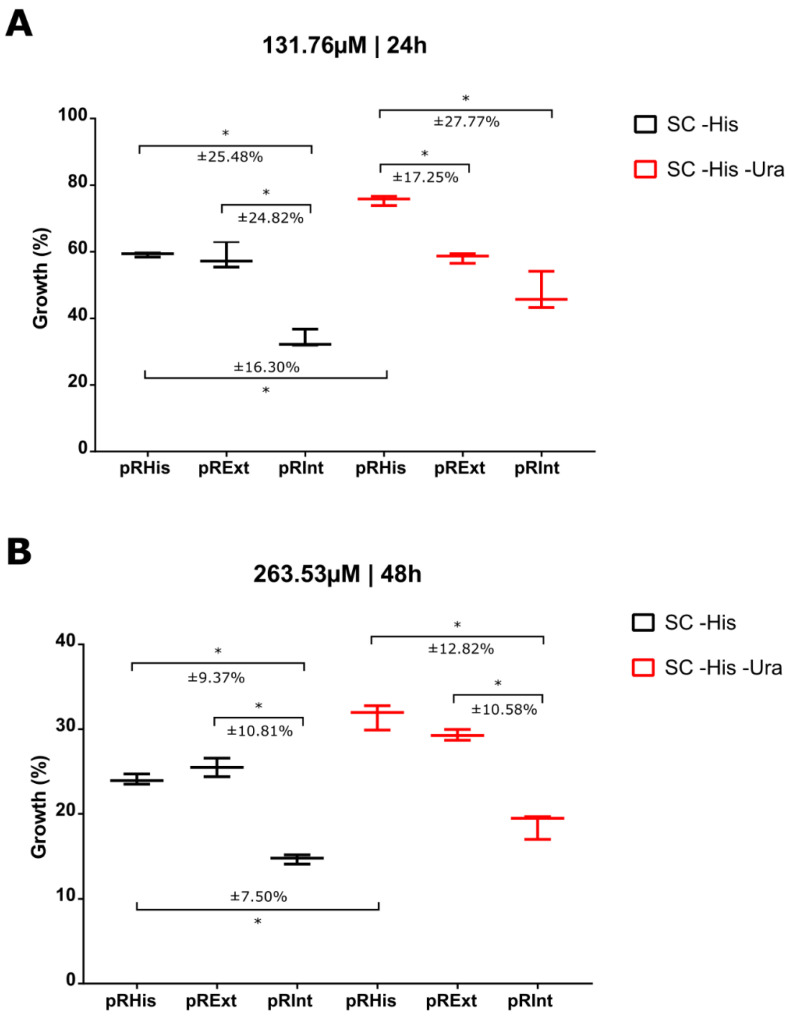
Impact of cisplatin on the growth of *S. cerevisiae* containing the CnePrp8i in the *URA3* gene (pRInt) in the SC -His and SC -His -Ura media. The growth was compared with *S. cerevisiae* controls (*URA3* gene without the CnePrp8i, pRHis and pRExt). Points of greatest growth difference between the constructs under cisplatin availability (**A**) at 131.76 μM after 24 h growth and (**B**) at 263.53 μM after 48 h growth. (* Tukey’s HSD test *p* < 0.001).

## Data Availability

All plasmid constructs will be made available upon request.

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
