# Peer review of "Cryptococcus neoformans Prp8 Intein: An In Vivo Target-Based Drug Screening System in Saccharomyces cerevisiae to Identify Protein Splicing Inhibitors and Explore Its Dynamics"

_jof, 2022, doi:10.3390/jof8080846_

Round 1

Reviewer 1 Report

The authors described development of an cell-based protein splicing assay using Saccharomyces cerevisiae to mimic Prp8 intein splicing in pathogenic fungus Cryptococcus neoformans. They inserted the gene encoding fungal Prp8 intein to the gene Ura3, which is essential for yeast cell survival in uracil-less medium. The authors successfully created such a yeast cell and characterized it using Cisplatin, a model intein splicing inhibitor. Overall the manuscript is technically sound. My only minor concern is that an unspliced Ura3-Prp8 intein should be detect for the mutant pRMut using western blot (fig 5).

Author Response

(Please see the attachment)

Response to Reviewer 1 Comments

Point 1: Overall the manuscript is technically sound. My only minor concern is that an unspliced Ura3-Prp8 intein should be detect for the mutant pRMut using western blot (fig 5).

Response 1: Thank you for your comment. Indeed, this is an important concerning in this work that we should clarify better to the readers.

We agree that the observation of an unspliced Ura3-Prp8i band, in western blot, would ideal to confirm the splicing inability of this mutated intein. However, this was a limitation of this system because: i) in SC -His -Ura medium, yeasts with this construction do not grow (as expected and shown in figure 3) and ii) in SC -His medium, the ura3 expression is low (since there is uracil in the medium), as evidenced in the graphic of figure 5B. In the discussion, lines 396-401, we added a comment on this concerning, suggesting that a stronger promoter would overcome this problem, since the ura3expression would become high enough (even in SC -His medium) to be detected in Western Blots.

Reviewer 2 Report

The manuscript by Fernandes JAL et al. presents an interesting study on the construction of an in vivo drug screening system in Saccharomyces cerevisiae targeting Prp8 intein of the pathogenic yeast Cryptococcus neoformans. To the authors' knowledge this is the first in vivo intein-targeting drug screening system built which could be utilised for the high throughput screening and identification of protein splicing inhibitors as potential antifungal drugs. Experiments were nicely performed and logical. Below are some comments for the authors' consideration:

Line 47: central nervous system (no capitalisation)

Line 69: 'task' instead of 'ask'

Line 70: filamentous fungi are also multicellular organisms?

Line 102: 'dermatophytes' instead of 'dermatomycoses'

Lines 132-133: It may be better to use His- & Ura- (superscript -) instead of -His & -Ura for easier understanding

Lines 141-142: How was this performed?

Line 205: Where was S. cerevisiae strain W3O3 obtained?

Figure 6: I am not an expert in statistics but I wonder why the authors used multiple t tests for pairwise statistical comparison instead of ANOVA. The construction of S. cerevisiae control (URA3 gene without the CnePrp8i, pRHis and pRExt) is not mentioned. Also, they was pRMut not included for comparison?

Line 305: It is said that cisplatin showed an IC50 of 201.65 μM to S. cerevisiae cells while a concentration of 131.76 μM of cisplatin is needed to inhibit S. cerevisiae growth after 24h (Line 307). What is the cytotoxicity to mammalian cells at this concentration (131.76 μM)? Would this affect the utility of this drug candidate as an potential antifungal drug if the drug at this concentration is cytotoxic to mammalian (especially human) cells?

Lines 358-361: The explanation provided here for the paint PCR band in non-transformant yeast is not sufficient. Even though there is a nucleotide substitution leading to an amino acid mutation so that the protein Ura3 become non-functional in W303, the mutated Ura3 gene is still expressed and RNA can still be expressed. This can be detected using RT-PCR but why would the PCR band become faint? Would this be related to the PCR primers? If so, why did the authors not used a more specific primer for detection?

Lines 371-376: What message(s) do the authors wish to express in this paragraph?

Line 381: Cu2+ instead of Cu+2 (also the charge should be indicated in superscript). Similarly for Zn+2 in line 382.

Guidelines for gene/protein nomenclatures should be better followed. (e.g. italics vs non-italics, captilised vs non-capitalised). Best practice should be followed so as to aid in readers' understanding.

Line 437: Data availability: It would be best if the various plasmid constructs can be deposited to public culture collections. If this is not feasible, they should at least be made available upon request. 

English language should be improved. The authors are suggested to seek help from a native English speaker.

Author Response

(Please see the attachment)

Response to Reviewer 2 Comments

Thank you for your comments.

Point 1: Line 47: central nervous system (no capitalization)

Response 1: the words were corrected, and capitalization was removed.

Point 2: Line 69: 'task' instead of 'ask'

Response 2: the word ask was replaced (corrected) with task.

Point 3: Line 70: filamentous fungi are also multicellular organisms?

Response 3: Indeed, fungi are considered multicellular, so to make it clear we replaced the term “multicellular” with “metazoan organisms”.

Point 4: Line 102: 'dermatophytes' instead of 'dermatomycoses'

Response 4: the word dermatomycoses was replaced with dermatophytes.

Point 5: Lines 132-133: It may be better to use His- & Ura- (superscript -) instead of -His & -Ura for easier understanding

Response 5: We used this nomenclature (SC-His-Ura, for ex.) because this is the way the drop-out media, from many companies (Takara, clonthec, etc) are named. Also, this is the nomenclature for drop-out media used in most yeast research. Furthermore, the use of superscript – after the words his and ura could be misunderstood as yeast genotypes.

Point 6: Lines 141-142: How was this performed?

Response 6: We detailed this with a new paragraph: “The S. cerevisiae Ura3 protein sequence was manually analyzed and advantageous possible intein insertion sites were identified by observing three key points: (i) biochemical similarities to the intein native insertion site, (ii) location in a non-rigid region (i.e. loop regions) and (iii) sites not directly related to the protein function” in lines 142-145.

Point 7: Line 205: Where was S. cerevisiae strain W3O3 obtained?

Response 7: S. cerevisiae strain W303 was obtained from the Laboratory of Molecular and Cellular Biology of Microorganisms, School of Pharmaceutical Sciences, UNESP, Araraquara, São Paulo, Brazil. Standard methods were used for its growth and manipulation. This information was added in lines 210-212.

Point 8: Figure 6: I am not an expert in statistics but I wonder why the authors used multiple t tests for pairwise statistical comparison instead of ANOVA. The construction of S. cerevisiae control (URA3 gene without the CnePrp8i, pRHis and pRExt) is not mentioned. Also, they was pRMut not included for comparison?

Response 8: Indeed ANOVA is more suitable for such analysis and we performed this statistical analysis again using this method. The significance of differences was maintained and did not change our conclusions. Part of the methodology section (lines 261-266) and also figure 6 were updated in the MS. Concerning the pRMut, it was not included because, as expected, it does not grow in SC -His -Ura medium. This was actually a limitation of the system, now discussed in lines 396-401. This mutated control was created with the main objective to demonstrate that the Ura3 protein with a retained intein (non-spliceable intein) is not functional (as shown in figure 3).

Point 9: Line 305: It is said that cisplatin showed an IC50 of 201.65 μM to S. cerevisiae cells while a concentration of 131.76 μM of cisplatin is needed to inhibit S. cerevisiae growth after 24h (Line 307). What is the cytotoxicity to mammalian cells at this concentration (131.76 μM)? Would this affect the utility of this drug candidate as an potential antifungal drug if the drug at this concentration is cytotoxic to mammalian (especially human) cells?

Response 9: Indeed Cisplatin is a cytotoxic compound. The cytotoxicity of Cisplatin for mammalian cells varies depending on cell type. According to Kumar and Tchounwou, (doi: 10.18632/oncotarget.5754) and Kaeidi et al (https://doi.org/10.3109/0886022X.2013.829406) its cytotoxicity is around 70-80 uM (about 35% - 50% of cytotoxicity). Therefore, the concentration of 131.76 uM is cytotoxic for mammalian cells, but Cisplatin was here used just to validate our system because it is one of the main compounds with a known inhibition effect on intein splicing at the time we created the system. Recently new compounds have been described, as discussed in the discussion section. However, the main objective of the article was to create a system that can be able to validate splicing inhibitors selected by in vitro high-throughput screening systems.

Point 10: Lines 358-361: The explanation provided here for the paint PCR band in non-transformant yeast is not sufficient. Even though there is a nucleotide substitution leading to an amino acid mutation so that the protein Ura3 become non-functional in W303, the mutated Ura3 gene is still expressed and RNA can still be expressed. This can be detected using RT-PCR but why would the PCR band become faint? Would this be related to the PCR primers? If so, why did the authors not used a more specific primer for detection?

Response 10: Indeed, this faint band was intriguing. We have checked the sequences of the primers and they are totally complementary to the ura3 gene (both from the plasmid and also from the W3O3 nuclear genome). The faint band of pRInt and pRMut, together with the ura3-prp8i (the larger band), in the same sample, indicated that the nuclear mutated ura3 gene is less expressed than the plasmid ura3. We clarified this in lines 382-387.

Point 11: Lines 371-376: What message(s) do the authors wish to express in this paragraph?

Response 11: we just wish that the reader understands that the splicing of the Prp8 intein does not occur at the RNA level. Also, we corrected the explanation about the faint band, as suggested in the previous comment (Lines 382-387).

Point 12: Line 381: Cu2+ instead of Cu+2 (also the charge should be indicated in superscript). Similarly for Zn+2 in line 382.

Response 12: We corrected the ions nomenclature.

 Point 13: Guidelines for gene/protein nomenclatures should be better followed. (e.g. italics vs non-italics, captilised vs non-capitalised). Best practice should be followed so as to aid in readers' understanding.

Response 13: We followed the genetic nomenclature for yeast. Gene names are in upper case and italicized (URA3, PRP8genes) and protein names have the first letter capitalized and non-italic (Ura3 protein, Prp8 intein).

 Point 14: Line 437: Data availability: It would be best if the various plasmid constructs can be deposited to public culture collections. If this is not feasible, they should at least be made available upon request. 

Response 14: We added in the text the information that “All the constructed plasmid may be available upon request” in the Data Availability Statement section (line 476).

Point 15: English language should be improved. The authors are suggested to seek help from a native English speaker.

Response 15: A native English speaker scientist, Dr. Pierre Mattar (OHRI and uOttawa) has kindly proofread our manuscript and we added him in the acknowledgments section.

Reviewer 3 Report

In this study, Lourenco Fernandes et al. have developed a novel drug screening system to assess the splicing of inteins in Cryptococcus that may help in the development of new mycosis treatment options. The manuscript addresses an interesting topic, is well designed and reads nicely.  

I would recommend to add a list of abbreviations because of the complexity of the terms used and to help the reader navigate. 

Also, limitations and setbacks could be discussed more, as well as strategies to overcome them and future prospects.

Author Response

(Please see the attachment)

Response to Reviewer 3 Comments

Point 1: I would recommend to add a list of abbreviations because of the complexity of the terms used and to help the reader navigate. Also, limitations and setbacks could be discussed more, as well as strategies to overcome them and future prospects.

Response 1: Thank you for your comment. Indeed, there are many abbreviations that, despite their meanings being addressed the first time they appear in the text, may confuse some readers. We added a list of abbreviations as suggested (line 489).

Concerning the limitations of in vivo screening systems, like ours, we had already mentioned in the discussion, that these systems do not isolate the target (intein, in this case) as the in vitro systems do, so that in vivo screening can be used for validating inhibitors discovered in vitro, as well as to check possible cytotoxicity of such compounds. Concerning the limitations specific to our system, using Ura3 as a non-native extein, we added, as also requested by Reviewer #1, a discussion about the non-detection of an unspliced band of Ura3-Prp8i precursor, in the western blot, for the pRMut construction (lines 396-401). In this discussion, we proposed the use of stronger promoters to overcome this problem.

Round 2

Reviewer 2 Report

The authors have addressed most comments from the previous round of review. Below are some additional suggestions for their consideration:

Line 65: 'and' need not be italicised.

Lines 95-96: Bacterial/prokaryotic gene names need to be italicised and are in lower case? Please check the rules for nomenclature.

Line 248: 'Protein analysis S. cerevisiae manipulation"?

Lines 308-309: Does the clause 'which demonstrate that ... ' describe 'intein-less controls exhibited an 845 bp PCR product'?  This may sound wierd here since these are 'intein-less controls' so CnePrp8i splicing is not expected at all.

Lines 307-313: The sentences here should be rewritten to convey the message more clearly. It may be useful to state which lanes the authors are referring to in the text.

Lines 326-335: The authors used anti-6xhis-tag antibody to perform

Line 357: exhibited

Lines 442-450: Ion nomenclature is still not corrected.

Author Response

(Please see the attachment)

Response to Reviewer 2 Comments (Round 2)

We thank Reviewer 2 for all the positive and constructive remarks. All the suggestions were accepted and included in a new version of this manuscript. All included modifications are highlighted in both .pdf and .docx files.

Point 1: Line 65: 'and' need not be italicised.

Response: “and” was corrected to a non italicised word, line 60.

Point 2: Lines 95-96: Bacterial/prokaryotic gene names need to be italicised and are in lower case? Please check the rules for nomenclature.

Response: radA and sufB was re-written using the correct nomenclature, lines 86 and 87.

Point 3: 'Protein analysis S. cerevisiae manipulation"?

Response: “'Protein analysis S. cerevisiae manipulation” was replaced with “Protein analysis”, line 247.

Point 4: Lines 308-309: Does the clause 'which demonstrate that ... ' describe 'intein-less controls exhibited an 845 bp PCR product'?  This may sound wierd here since these are 'intein-less controls' so CnePrp8i splicing is not expected at all.

Response: The clause “'which demonstrate that […]” refers to “1400 bp PCR product”. We rewrote in lines 307-319. We hope it is clearer for the readers now.

Point 5: Lines 307-313: The sentences here should be rewritten to convey the message more clearly. It may be useful to state which lanes the authors are referring to in the text.

Response: The sentences were rewritten in lines 307-319. We hope it is clearer for the readers now.

Point 6: Lines 326-335: The authors used anti-6xhis-tag antibody to perform

Response: The sentences were rewritten in lines 334-335. We hope it is clearer for the readers now.

Point 7: Line 357: exhibited

Response: the word “exhibit” was replaced by “exhibited”, line 373.

Point 8: Lines 442-450: Ion nomenclature is still not corrected.

Response: the ions were corrected to Cu2+ and Zn2+, lines 462 and 463.
